# Changes in Body Composition and Anthropomorphic Measurements in Children Participating in Swimming and Non-Swimming Activities

**DOI:** 10.3390/children8070529

**Published:** 2021-06-22

**Authors:** Grzegorz Bielec, Anna Gozdziejewska, Piotr Makar

**Affiliations:** 1Department of Tourism, Recreation and Ecology, University of Warmia and Mazury in Olsztyn, ul. Oczapowskiego 5, 10-719 Olsztyn, Poland; gozdzik@uwm.edu.pl; 2Department of Swimming, University School of Physical Education and Sport in Gdansk, ul. Gorskiego 1, 80-336 Gdansk, Poland; piotrmakar@wp.pl

**Keywords:** body composition, body mass index, physical activity, pubescence, training

## Abstract

Background. Physical activity is a well-known means of obesity prevention, but the relationship between exercise frequency and body composition in children has not been thoroughly investigated. Objective: The aim of this study was to compare the body composition of children aged 11–12 who regularly performed swimming and other sports as an organized extra-curricular physical activity for a 12-week period. Methods: The study included 46 students who attended swimming classes and 42 students who participated in training activities in other sports, including, but not limited to, football, basketball and athletics. Body height and body composition were measured using a Tanita BC 418 MA analyzer. The students individually reported their rate of perceived exertion during training using the Pictorial Children’s Effort Rating Table PCERT scale. Results: The weekly volume of training was substantially higher in the group of swimmers than in that playing other sports (12.3 h/week vs. 5.2 h/week, *p* < 0.01). After 12 weeks of training, body height and weight significantly increased in both groups (*p* < 0.001). However, the BMI value and adipose tissue content only increased in the group of non-swimmers. Swimmers perceived greater exertion during training than non-swimmers (7.1 vs. 5.8 on the PCERT scale, *p* < 0.01). Conclusions: In early pubescent children, engaging in vigorous exercise such as swimming for at least 10 h a week may restrain the growth of adipose tissue. However, the variety of exercises that are typical of team sports, if performed for no more than 5 h a week, may be insufficient to restrain adipose tissue growth.

## 1. Introduction

Regular physical activity, if performed at a sufficient intensity and with adequate frequency, has a positive effect on cardiovascular, mental and musculoskeletal health, and adiposity [1]. In children and adolescents, the most common result of an insufficient amount of physical activity and an improper diet is being overweight or obese, whereas intensified physical activity reduces body fat content in prepubertal children [2,3,4]. The World Health Organization (WHO) states that children should be encouraged to participate in various kinds of physical activity to improve cardiorespiratory and muscle fitness and reduce symptoms of anxiety and depression [5]. Following WHO suggestions, American, African and European governmental organizations have presented guidelines on the recommended daily amount of physical activity for children according to regional context [6,7]. In most countries, it is recommended that children engage in moderate-to-vigorous intensity exercise for at least 60 min a day. The energy costs of different sports activities performed by children can vary; for example, gymnastics demands 4 METs (metabolic units), basketball 8.2 METs, soccer 8.8 METs, and swimming from 8.4 to 11.6 METs depending on the velocity of swimming [8]. Additionally, children should limit to 2 h a day the amount of time spent using a computer, playing video games and watching TV.

The level of compliance with the recommended amounts of physical activity in schoolchildren differs widely. In Latin America, 2 h of recreational screen-time is frequently exceeded by school-age children [9]. Moreover, only 15% of Latin American adolescents undertake 60 min of moderate-to-vigorous exercise per day [10]. In the United States, children and adolescents aged 6–17 are also insufficiently active, and only 24% of them engage in 60 min of exercise every day [11]. In South Africa, on the other hand, nearly 70% of 8–14-year-old children undertake moderate-to-vigorous physical activity [12]. In this regard, European countries are very diverse. Swiss children dedicate only a small percentage of their daytime to running or jumping, instead spending 75% of the time in a still position (mostly sitting) [13], whereas in a group of Estonian children, 11% of the children met the desired criteria for physical activity, in a group of British children, 70% of the subjects met the criteria [14,15]. A different study involving 9-year-old British children attending ad hoc and scheduled sports activities with various weekly frequencies led researchers to conclude that more frequent participation in sports activities (up to five times a week) substantially contributes to compliance with the recommended amounts of physical activity [16].

Extracurricular sports activities are usually organized by educational institutions, sports clubs and other organizations that are supervised by adults. Children may often participate in these activities for free, but in sports clubs, a membership fee may be required. Participation in an organized extra-curricular sport has a stronger effect on developmental factors, such as proper body mass, BMI and academic achievements, than spontaneous physical activity [17]. Moreover, organized physical activity can be defined more precisely in terms of training loads than spontaneous, self-organized physical activity.

To our knowledge, no research has been undertaken so far to examine anthropometric differences among children who participate in extra-curricular physical activities of various volumes (e.g., 30 min, 60 min or 90 min) and frequencies (e.g., once a week, three times per week or five times per week). Therefore, the objective of this study was to determine the relationships between body composition and regular physical activity of various volumes (5–12 h/week) performed over a 12-week period of time by 11- to 12-year-old children.

## 2. Materials and Methods

### 2.1. Participants

We aimed to create a study group with children with similar socioeconomic backgrounds and similar daily routines due to attendance at the same public school. Therefore, we chose a quasi-experimental, two-group design that used non-random sampling. The study involved 100 students attending a state-funded primary school, which serves approximately 600 students from a range of socioeconomic backgrounds in the city of Olsztyn, Poland. Students at this school attend compulsory physical education classes for 3 h a week. The students were divided into two groups: 50 children (including 21 girls) who regularly participated in swimming workouts in addition to compulsory physical education, and 50 children (including 22 girls) who regularly participated in other forms of extra-curricular physical activity (details on duration and frequency of activity are presented in Results). Children participating in non-swimming sports were grouped together for two reasons: (i) the number of children engaged in each non-swimming discipline (e.g., in basketball) was too small to create separate groups; (ii) the children in the group of non-swimmers constitute all of the school’s pupils aged 11–12 years who participated in any organized extra-curricular sports activities other than competitive swimming. The examined children of both groups did not differ substantially in terms of socioeconomic status, i.e., they lived in the same residential area, were being raised by two parents of middle-to-higher education background and had 1–2 siblings. The information on the children’s participation in these extra-curricular sporting activities was collected by physical education teachers during school classes. The average age in the group of swimmers was 11.7, whereas that in the group of non-swimmers was 11.9 (non-significant difference).

### 2.2. Measurements

In September 2018, the first anthropometric measurements were taken in both groups. Body height was measured to one decimal place (0.1 cm) by means of a Seca 216 stadiometer (Seca GmbH, Hamburg, Germany). Body mass, BMI and body fat percentage were estimated using a Tanita BC 418 MA analyzer (Tanita Corp.,Tokyo, Japan). The measurements were conducted in the school nurse’s office between 9:00 a.m. and 11:00 a.m. The children were dressed in light clothes without shoes. Each measurement of body height and body composition took about 12 min. The measurements were taken twice and the coefficient of variation (CV) was calculated for each pair of measurements. The range of CV values was from 2.2 to 3.2. The children conducted a self-assessment of the stage of their sexual development at home, in the presence of their parents, on the basis of the Tanner scale [18]. In September 2018, children from both groups were acquainted with the Pictorial Children’s Effort Rating Table (PCERT) scale of the subjective assessment of physical exertion [19]. The children were asked to regularly record their assessment of intensity of exertion immediately after their training session every week. The children from the group of swimmers continued to record this every Tuesday for 12 weeks. Non-swimmers recorded the data every Tuesday or Wednesday for the same duration, depending on the day their training was scheduled. In October 2018, both groups were surveyed on their eating habits. The reliability and accuracy of the questionnaire used had been validated by the studies conducted in the past on children aged 10–13 [20]. Eating habits did not differ significantly between groups. In December 2018, after 12 weeks from the anthropometric measurements, follow-up measurements were performed according to the same procedure. As some of the children had been absent from the sports activities (20% or more absences within 12 weeks), the results of 46 children (including 19 girls) from the swimmers and 42 children (including 19 girls) from the non-swimmers were taken into consideration.

### 2.3. Ethical Clearance

The study was conducted in accordance with the Declaration of Helsinki. The head teacher of the school gave permission for conducting the tests, and the parents of each child provided written consent for their children to participate in the study. Informed consent was also obtained from all children participating in the study. Finally, the Ethics Committee of the University of Warmia and Mazury in Olsztyn approved the study (KB-8/17).

### 2.4. Statistics

The statistical calculations were performed using Statistica 12.0 software (StatSoft, Tulsa, OK, USA). According to the Shapiro–Wilk test, the distribution of the data was normal. The results of the first and second anthropometric measurements were compared using a *t*-test. Statistical significance was set at *p* < 0.05. The statistical power of the comparisons varied from 0.32 to 0.46.

## 3. Results

Table 1 presents the weekly volume of training in both groups and their self-assessments of their physical exercise intensity. The children from the group of non-swimmers declared their participation in the following sports activities: football (9), basketball (6), swimming (6), athletics (4), martial arts (4), dance classes (4), fencing (3) and table tennis (3). To clarify, swimming activities by children in the non-swimmers group differed from the training sessions of those in the swimmers group. Non-swimmers participated in three 45 min classes per week, focusing on technical aspects and usually covering 700–800 m during each class. Swimmers participated in eight to ten 90 min workouts per week, covering 3.6–4.2 km per workout. In individual cases among the non-swimmers, the children stated that they participated in other sports on a regular basis, e.g., acrobatics, horse riding, wrestling and roller skating. The students from the group of swimmers participated in training sessions considerably more frequently. In addition, they perceived their exertion in these sessions as substantially more difficult than that of their peers from the group of non-swimmers. Swimmers mostly perceived their exertion as “hard”, whereas non-swimmers usually expressed their effort as “starting to get hard” and “getting quite hard”.

Figure 1, Figure 2, Figure 3 and Figure 4 present the results of the anthropometric measurements taken in September 2018 and December 2018. After 12 weeks, a noticeable increase in body height and weight was observed in both groups. However, a statistically significant increase in BMI value and body fat percentage was noted only in the group of non-swimmers. The correlation between BMI and body fat percentage was strong both in girls (*r* = 0.99) and in boys (*r* = 0.95).

## 4. Discussion

The aim of our study was to compare the effects of swimming and participating in other sports on the body composition of 11- to 12-year-old children. Our results indicate that, whereas the BMI and body fat content of swimmers may not change significantly over 12 weeks, they can increase significantly in children participating in other sports. It should be noted, however, that the indicators in both groups were within the national norms for this age group [21]. As in other studies [22,23], the correlation between body mass index and body fat percentage was strong in boys and girls (*r* > 0.95). Moreover, on average, swimmers exercised 12 h per week, and non-swimmers exercised for 5 h per week.

The lack of a significant change in the BMI and adipose tissue content of the swimmers might be due to the mostly aerobic nature of their activities. Laframboise and de Graauw [24] completed an analysis of the literature on the relationships between aerobic exercise and overweight schoolchildren. According to them, aerobic exercise might be recommended for children as a means of reducing body mass. The findings from our research are consistent with the remarks made by Ness et al. [25]. Those authors took anthropometric measurements and analyzed the physical activity of 5500 British 12-year-old children over a 3-day period. They came to the conclusion that unstructured physical activity influences body mass and BMI to a smaller extent than structured moderate-to-vigorous intensity exercise. Similarly, beneficial changes in body composition (i.e., fat mass and free fat mass) were larger in Spanish 11-year-olds that performed vigorous exercise for 12 weeks than in those that performed moderate exercise over this time period [26].

We believe that the weekly number of training sessions in swimmers and non-swimmers may also have affected the results of their body composition. Similar conclusions can be found in the literature. For example, in 10-year-old Brazilian children, the amount of vigorous physical activity taken during 1 week was inversely correlated with their BMI and body fat percentage [27]. Similarly, Pastuszak et al. [28] monitored sports-school students and non sports-school students for a period of over 3 years. Sports-school students participated in 15–25 h of sports activities a week, whereas non sports-school students participated in 4 h of physical education classes at school and 4 h of extra-curricular sports activities a week. Compared to the control group, the sports-school students had lower body mass and BMI values, and lower body fat percentages. In contrast, Kubusiak-Slonina et al. [29] did not find any statistically significant correlations between the BMI and the level of physical activity of 92 12-year-old Polish children. Similarly, BMI and body fat values did not differ significantly between 10-year-old South African children involved in team sports and their peers not participating in extra-curricular sports activities [30].

In the present study, the body composition indicators of the swimmers were very similar to those of swimmers from other European countries [31,32]. It is noteworthy that, in our study, the BMI values of swimmers and non-swimmers were within the national population norms [21]. In other words, the children examined in this study displayed ratios of height to body mass that are typical for their age in their regional and national context. However, studies in non-European contexts have reported noticeable differences in anthropometrics between children of different nationalities who engage in a similar number of training sessions per week with similar training loads [33,34].

The effects of maturation may also have affected the results presented here, particularly with regard to the girls. On the one hand, children involved in regular strenuous swimming workouts may exhibit delayed puberty and not put on as much body fat as children participating in sports requiring less intensive training [35]. For example, Damsgaard et al. [36] found that early pubescent female swimmers had a sum of skinfolds that was lower than that of handball players but higher than that of gymnasts. On the other hand, Opstoel et al. [37] reported that prepubertal swimmers had higher values of body fat and BMI than their peers engaged in athletics, ball sports, gymnastics and dance. This is similar to what we observed, as our swimmers also had greater BMI values and body fat percentage at the baseline than their non-swimming peers. We believe that this could be due to the early phase of the swimmers’ training phase. The first measurements were taken in mid-September, just 2 weeks after starting the new school-year season. At that time, the swimmers were not training at high intensities or high volumes, and thus their training sessions were not physiologically demanding.

The results obtained here, based on the PCERT scale, indicate that the swimmers perceived greater mean exertion during training than their peers involved in other sports (swimmers, 7.1; non-swimmers, 5.8). These differences may be caused by physiological factors. A study by Yelling et al. [19] indicated that 11–12-year-old girls and boys had an average heart rate of 170 beats per minute at stage 5 and 180 beats per minute at stage 7 on the PCERT scale, respectively. It is also possible that this perception is due to psychological factors. The swimmers started training every day at 6:00 a.m. and attended a second session in the afternoon three to five times a week. The non-swimmers usually started their exercise sessions in the afternoon and did not attend a second exercise session.

### Strengths and Limitations

This study makes a contribution to the literature by providing a comparison of the body composition of prepubescent children participating in different sports. Whereas the literature contains many comparisons of athletes and non-athletes in this regard, we are aware of only one other comparison of the body composition of prepubescent swimmers and that of children of the same age participating in other sports [37].

There are several limitations that should be kept in mind while assessing the results of this study. First, our study contained fewer participants than those in most sport science studies because we used only children who attended the same school. On the one hand, this helped to reduce the confounding effects of socioeconomic status and daily routine on our results. Moreover, because all of these children ate their midday meal in the school canteen, their diets would have been more similar than otherwise. On the other hand, due to the relatively small sample size, our study may have lacked the statistical power to detect smaller differences between groups. Although the power is low, we provide valuable data which can be combined with that of other studies via statistical meta-analysis to provide more precise estimates and higher power, as is commonly done in medical studies of children [38]. Second, heart-rate monitors were not used to measuring training intensity in the groups of swimmers and non-swimmers; thus, it was not possible to collect objective data on exercise intensity. Therefore, we were not able to verify whether the swimmers’ workouts were indeed more intense than the training sessions performed by subjects in the other sport disciplines. Third, our analysis of body composition focused on adipose tissue content, which in our view, is the most easily comparable indicator of risk. We considered the remaining variables (e.g., body height or body mass index) to be of less importance for the issue in question. Finally, we decided not to create a control group consisting of children not involved in structured extracurricular sports activities, as many studies have presented significant differences in body composition between physically active children and their inactive peers [39,40,41]. We also decided not to present the findings from our survey of nutritional habits, because the substantial differences between swimmers and non-swimmers were not revealed. Moreover, the literature provides examples of similar eating habits of youth swimmers and non-swimmers [42].

## 5. Conclusions

Our results indicate that physical activity perceived as “hard” (7 or more on the PCERT scale) that is engaged in for at least 10 h per week may stop the growth of body fat in prepubescent children. In contrast, activity perceived as “starting to get hard” (5 or less on the PCERT scale) performed for 5 h a week may not be sufficient to stop the growth of body fat in children of this age. Importantly, however, adipose tissue content in both groups remained within the national norms for children at this age. It would be interesting to independently confirm these results with a larger sample of children. Moreover, a confirmatory study could be conducted over a longer period of time to see if the trend of an increasing content of adipose tissue in moderately active children persists.

## Figures and Tables

**Figure 1 children-08-00529-f001:**
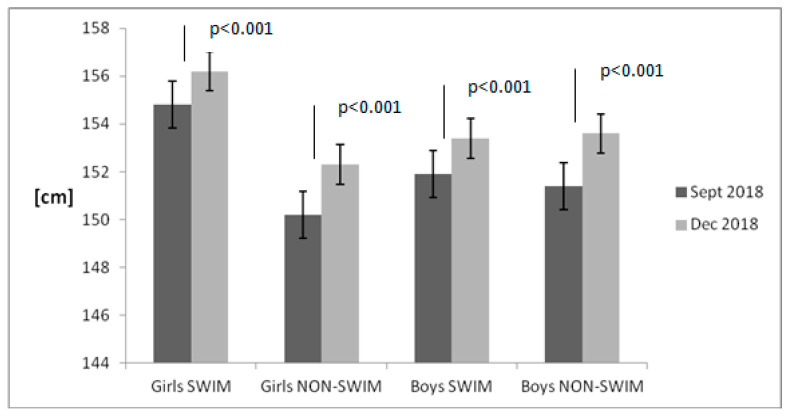
Body height changes in the examined groups of children. Data are presented as mean value and standard error. The non-swimmers group consisted of children participating in team sports (soccer, basketball), athletics, martial arts and dance.

**Figure 2 children-08-00529-f002:**
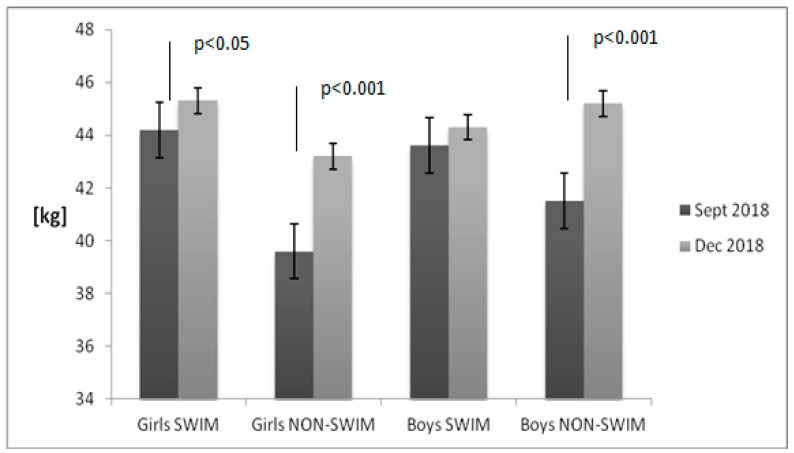
Body mass changes in the examined groups of children. Data are presented as mean value and standard error. The non-swimmers group consisted of children participating in team sports (soccer, basketball), athletics, martial arts and dance.

**Figure 3 children-08-00529-f003:**
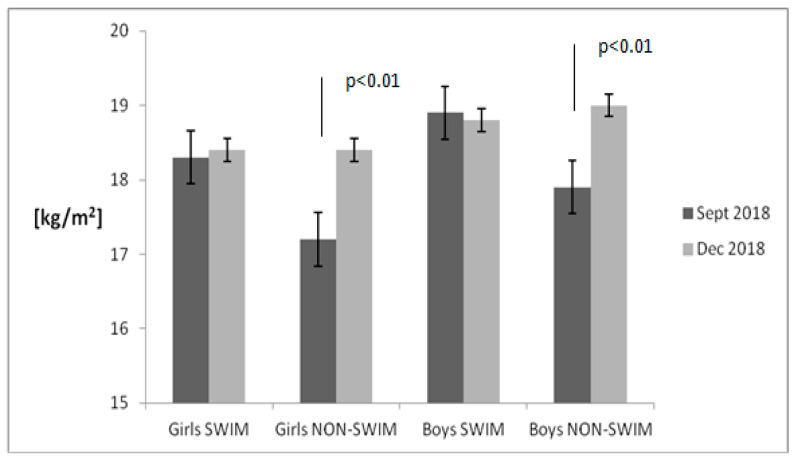
Body mass index (BMI) changes in the examined groups of children. Data are presented as mean value and standard error. The non-swimmers group consisted of children participating in team sports (soccer, basketball), athletics, martial arts and dance.

**Figure 4 children-08-00529-f004:**
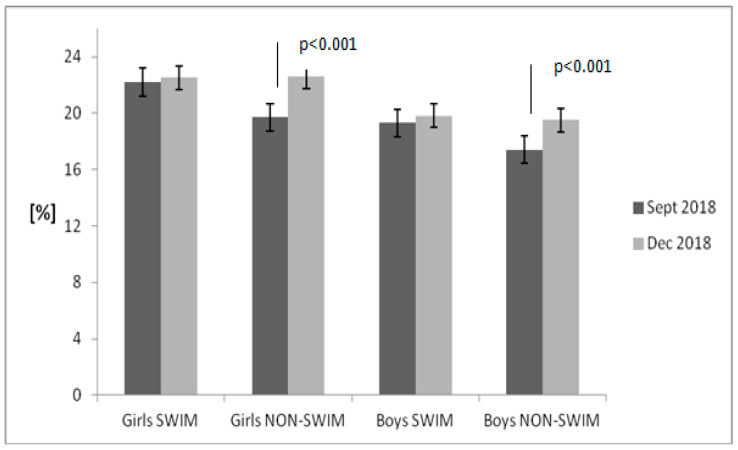
Body fat percentage changes in the examined groups of children. Data are presented as mean value and standard error. The non-swimmers group consisted of children participating in team sports (soccer, basketball), athletics, martial arts and dance.

**Table 1 children-08-00529-t001:** Training loads, puberty stage and rate of perceived exertion in the examined groups.

	Number of Training Sessions per Week Mean (SD)	Number of Training Hours per Week Mean (SD)	Puberty Stage (Tanner Scale) Mean (SD)	Rate of Perceived Exertion (PCERT Scale) Mean (SD)
Swimmers (*n* = 46)	8.2 (1.4) **	12.3 (2.2) **	2.0 (0.6)	7.1 (0.7) **
Non-swimmers (*n* = 42)	3.5 (0.9)	5.2 (1.8)	2.0 (0.7)	5.8 (0.8)

** *p* < 0.01 compared with non-swimmers group; SD—standard deviation; PCERT—Pictorial Children’s Effort Rating Table.

## Data Availability

The data presented in this study are available on request from the corresponding author.

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
