# Peer review of "Changes in Body Composition and Anthropomorphic Measurements in Children Participating in Swimming and Non-Swimming Activities"

_children, 2021, doi:10.3390/children8070529_

Round 1
Reviewer 1 Report
Page 1:
Line 10-12: This paper also did not fully investigate the relation between the exercise frequency and the body composition. So the sentence need to be corrected to reveal or include the scope of this work.
Page 2:
Line 60-66:
This research might contain the unique studies but there are tons of researches for the relation between the exercises and BMI and/or Body Compositions. Should we compare the extra-curricular activities and the normal exercises and say the extra-curricular activities are unique?
Line 75-82:
Is there any reason why the group is separated swimmers and non-swimmers?
In the contents, we also need to see how other activities such as basketball and soccer, etc. are intensive comparing with swimming. The general references of calories spending for all activities mentioned in this study will be helpful to understand the energy consumption of the extra-curricular activities.
Line 87:
Please explain more details regarding Tanita BC 418 MA analyzer. (spec. measurement method and frequencies, times, etc.)
Page 3 & 4:
Table 1 and 2 can be expressed by graphs and it may give more visual information to the readers.
Page 4:
Line 145:
Please specify why the extra-curricular activities are expressed to swimming and other activities.
If possible, quantify the effects of activities from the references.
Page 5:
Please corelate BMI and Body fat % and analyze the results.
Also, if you have the ratio of ECF and ICF, it will be very efficient to explain the effects of exercise.
Author Response
On behalf of authors of the article, I would like to thank you for objective and thorough review. Proper comments were taken into account and with no doubts there were contribute to improve the scientific level of the paper. After receiving reviews authors intention was to incorporate changes (there were marked with red color in the text).

Reviewer 2 Report
Introduction appears to be missing more details. Introduction briefly mentions physical activity and screentime of children from across the world, but could possibly dive deeper into the differences in physical activity from physical education, extracurriculars, etc.
In the methods, there is little data about the participants. Body mass, height/weight, BMI could all vary not just because of age. Methods mention all participants were from one school, but the demographics of that school were not mentioned. This information could also impact the results. Methods mention they asked the participants about their eating habits, but eating habits were not mentioned in the introduction, nor in the results. Could simply asking about nutrition mid-test influenced the data? Methods also report asking participants to rate themselves on the Tanner scale, but this is not mentioned in the results. The simple differences in Tanner scale could have impacted the results as well.
Results section is where clean-up could be used. The tables are very difficult to read, follow and understand just based on formatting. What is the power of this study? As mentioned below, a limitation is the number of participants. The results also indicate the non-swimmers participated in three, 45-minute sessions a week, whereas the swimmers participated in 8-10, 90-minute sessions a week. That alone right there could be reasons why swimmers have a lower mass than the non-swimmers, which is mentioned in the discussion, but it's difficult to not conclude this is the reason there are significant differences between the two sets of groups.
Author Response

(The authors gave the same response as above.)
